# Arginine-Rich Cell-Penetrating Peptide-Mediated Transduction of Mouse Nasal Cells with FOXP3 Protein Alleviates Allergic Rhinitis

**DOI:** 10.3390/pharmaceutics15061770

**Published:** 2023-06-19

**Authors:** Toru Miwa, Yumi Takemiya, Kazuki Amesara, Hiroko Kawai, Yuichi Teranishi

**Affiliations:** 1Department of Otolaryngology, Osaka Metropolitan University, Osaka 5458585, Japan; takemiya@ent-ocu.com (Y.T.); amesara@ent-ocu.com (K.A.); kawai@ent-ocu.com (H.K.); teranishi@ent-ocu.com (Y.T.); 2Department of Otolaryngology-Head and Neck Surgery, Kyoto University, Kyoto 6577575, Japan

**Keywords:** allergic rhinitis, cell-penetrating peptide, FOXP3, ovalbumin-specific IgE transgenic mice

## Abstract

Intranasal corticosteroids are effective medications against allergic rhinitis (AR). However, mucociliary clearance promptly eliminates these drugs from the nasal cavity and delays their onset of action. Therefore, a faster, longer-lasting therapeutic effect on the nasal mucosa is required to enhance the efficacy of AR management. Our previous study showed that polyarginine, a cell-penetrating peptide, can deliver cargo to nasal cells; moreover, polyarginine-mediated cell-nonspecific protein transduction into the nasal epithelium exhibited high transfection efficiency with minimal cytotoxicity. In this study, poly-arginine-fused forkhead box P3 (FOXP3) protein, the “master transcriptional regulator” of regulatory T cells (Tregs), was administered into the bilateral nasal cavities of the ovalbumin (OVA)-immunoglobulin E mouse model of AR. The effects of these proteins on AR following OVA administration were investigated using histopathological, nasal symptom, flow cytometry, and cytokine dot blot analyses. Polyarginine-mediated FOXP3 protein transduction induced Treg-like cell generation in the nasal epithelium and allergen tolerance. Overall, this study proposes FOXP3 activation-mediated Treg induction as a novel and potential therapeutic strategy for AR, providing a potential alternative to conventional intranasal drug application for nasal drug delivery.

## 1. Introduction

Allergic rhinitis (AR) is an inflammatory disease triggered by an immunoglobulin E (IgE)-mediated atopic response. It is characterized by nasal mucosal inflammation and manifests as rhinorrhea, sneezing, itching, and nasal congestion [1,2]. The rising incidence of AR poses serious public health issues [1,2]. Various therapeutic modalities, including antihistamines, steroids, montelukast inhibitors, and immunotherapy are used to treat AR. Among these, intranasal corticosteroids are the most effective and are regarded as the treatment of choice. Nasal administration is primarily suitable for potent drugs, as only a limited volume can be dropped or sprayed into the nasal cavity. Continuous and frequent administration of drugs may be necessary to achieve long-term effects in the nasal epithelium because drugs are rapidly eliminated from the nasal cavity by mucociliary clearance [3]. The physicochemical properties of a drug, as well as its potency and molecular size, are crucial factors in formulating a solution for nasal delivery. Numerous studies have been conducted to evaluate the appropriate types of drugs and methods for improving intranasal administration [4,5]. However, few studies have examined nasal administration of large proteins and peptides, as small substrates are more easily delivered into the brain than large peptides [6]. Thus, a novel and alternative method for delivering large proteins and peptides is needed to adjust the drug absorption time, concentration, and contact time between the drug and nasal mucosa, and prevent removal of the drug by mucociliary clearance. Among intranasal corticosteroids, mometasone furoate (MF) is a topically effective new-generation intranasal corticosteroid with low systemic absorption and a high affinity for glucocorticoid receptors [2,7]. However, the slow breakdown of MF in suspension delays the commencement of its activity and enables its mucociliary clearance from the nasal cavity [8]. This postpones the onset of its effects and hastens its elimination.

A possible solution for this issue is the use of cell-penetrating peptides (CPPs), a diverse family of short peptides that can enter several mammalian cell types [9,10]. Various macromolecules can be attached to these peptides and subsequently internalized. Cargo molecules that penetrate cells maintain their biological activities [9,10]. Among the different CPPs, arginine-rich CPPs have been most widely studied. An example is the protein transduction domain of the HIV-type 1 TAT protein, which contains a high proportion of arginine and lysine residues that are responsible for its ability to penetrate the plasma membrane [9,10] when added to culture media. Moreover, simple polyarginine peptides with an optimal length of 9–11 residues show significantly higher cell penetration rates than TAT [11]. Extensive research has revealed that arginine-rich CPPs [11] contain a high proportion of lysine and arginine residues, which enable penetration across the plasma membrane [10,11]. Our earlier studies have demonstrated that a peptide containing nine arginine residues (9R) efficiently transported enhanced green fluorescent protein (EGFP) into the nasal epithelium and brain of mice [12] without causing any damage to the nasal mucosa or brain. The protein was also transported into the developing inner ears of mice [13] and adult guinea pigs [14] without impairing their auditory or vestibular function. Reports have demonstrated the efficacy of the 9R peptide in delivering cargo into mouse nasal mucosa and maintaining prolonged contact with the nasal mucosa, bypassing drainage by mucociliary clearance. Polyarginine-mediated protein transduction in cells results in EGFP signals that last for 12–96 h, whereas EGFP signals without polyarginine-induced transduction last for only 12 h [12].

Sublingual immunotherapy (SLIT) is another AR-therapeutic strategy that is clinically used to treat type 1 allergies [15]. SLIT induces allergen tolerance, presumably through the reprogramming of allergen-specific T helper 2 (Th2) cells to Th1 cells and the production of peripheral CD4^+^ regulatory T cells (Tregs). Because the transfer of SLIT-induced Tregs provides tolerance against antigens, production of Tregs is necessary for SLIT efficacy [15,16]. Forkhead box P3 (*FOXP3*) is widely recognized as a Treg marker and is designated as the “master transcriptional regulator” of Tregs [17,18]. *FOXP3* expression may stimulate the differentiation of naïve and memory CD4^+^ T cells into Tregs [19]. Therefore, *FOXP3*-mediated Treg induction is a potential therapeutic strategy for AR. To date, cell-based therapy such as adoptive cell transfer of engineered Tregs has provided an effective therapeutic alternative to combat autoimmune diseases. However, no studies have reported directly on the application of the *FOXP3* gene or protein for treating AR.

The purpose of the current investigation was to explore the feasibility of 9R-mediated FOXP3 protein transduction into the nasal epithelium for therapeutic application in AR.

## 2. Materials and Methods

### 2.1. Recombinant FOXP3 and 9R Peptide-Conjugated FOXP3

Plasmids encoding *FOXP3* and *9R*-conjugated *FOXP3* (*FOXP3-9R*) were constructed for attachment of 9R to the C-terminus of FOXP3 protein (Appendix A). The plasmids were transfected into BL21 (DE3) competent *Escherichia coli* cells (New England Biolabs, Ipswich, MA, USA) [13,20]. Protein expression was stimulated using 0.1 mM isopropyl 1-thio-b-d-galactopyranoside. The expressed proteins were purified using a Ni-nitrilotriacetic acid–agarose column (Invitrogen, Carlsbad, CA, USA), and dialyzed with phosphate-buffered saline (PBS). Sodium dodecyl sulfate–polyacrylamide gel electrophoresis (SDS-PAGE) was performed for determining the molecular weight of FOXP3-9R (Appendix A). The concentration of FOXP3 and FOXP3-9R protein was 0.3 mg/mL and 0.2 mg/mL, respectively. We diluted them using PBS to obtain the same concentration (0.1 mg/mL). The proteins were then stored at −80 °C.

### 2.2. Immunoglobulin E (IgE) Transgenic Mice

Six-week-old male ovalbumin (OVA)-specific IgE transgenic mice (“BALB/cA-Tg(IgE-H01-4)Rin Tg(IgE-kL01-4)Rin/Jcl”) were obtained from CLEA Japan Inc. (Tokyo, Japan). OVA-IgE mice are genetically modified BALB/c mice that exhibit both acute and chronic allergic reactions with persistent production of IgE upon intraperitoneal administration of egg albumin [21,22]. The animals were kept in a climate-controlled space maintained at 25 °C and 50% relative humidity. All mice were fed a standard commercial pellet diet and had unlimited access to water. The Kitano Hospital “Committee on the Use and Care of Animals” approved all animal studies, which were conducted as per recognized veterinary standards (protocol number: A1900003). Euthanasia was performed via cervical dislocation, ensuring minimal distress and pain to the animals.

### 2.3. Protein Administration into the Mouse Nasal Cavity

FOXP3-9R and FOXP3 (control) proteins (10 µL aliquots, 0.1 mg/mL) were administered into the bilateral nasal cavities of the animals (5 mice/group) using a micropipette (“QSP, l0 µL filter tips; Thermo Fisher Scientific, Waltham, MA, USA). Before nasal administration, the micropipette tip was positioned 1–2 mm from the entrance of the nasal cavity while mice were awake in the supine position. After nasal treatment, mice were placed in a prone posture for 30 min. We divided the OVA-IgE mice into 6 groups: FOXP3 96 h before OVA (FOXP3 injected 96 h before OVA administration), FOXP3-9R 96 h before OVA (FOXP3-9R injected 96 h before OVA administration), FOXP3 24 h before OVA (FOXP3 injected 24 h before OVA administration), FOXP3-9R 24 h before OVA (FOXP3-9R injected 24 h before OVA administration), FOXP3 immediately after OVA (FOXP3 injected immediately after OVA administration), and FOXP3-9R immediately after OVA groups (FOXP3-9R injected immediately after OVA administration). An overview of the experimental procedure is illustrated in Figure 1.

### 2.4. Assessment of Nasal Symptoms

Nasal itching in OVA-IgE mice was investigated to assess the allergic reaction induced by intraperitoneal administration of OVA. Following FOXP3-9R, FOXP3, or saline administration, mice were observed for itching for 10 min, after a 10 min adaptation period at each time point [23], on days 1 (3 h, 12 h), 2 (24 h), and 3 (48 h) (Figure 1).

### 2.5. Immunohistochemistry

In groups injected with FOXP3-9R and FOXP3 24 h before OVA administration, a total of 10 mice (*n* = 5 per time point) were euthanized 3 h after OVA administration; none of the animals died before this point (Figure 1). This endpoint was selected to span the expected periods of maximum nasal symptoms, to determine any detrimental effects on the nasal mucosa resulting from the intranasal administration of FOXP3-9R and FOXP3. Mouse heads were dissected to perform a nasal examination, after which the dissected heads were fixed in 4% paraformaldehyde in PBS at 4 °C for 24 h. These tissues were then aligned, frozen in dry ice, and kept at −80 °C before sectioning. For cryostat sectioning, nasal lesions were embedded in an optimal cutting temperature compound (“Sakura Finetek Japan Co., Ltd., Tokyo, Japan”) and sectioned serially at 12 μm thickness. The cryostat sections were then washed twice with PBS (5 min/wash), and incubated with fluorescein isothiocyanate (FITC)-conjugated mouse anti-CD4 (RM4-5) antibody (Thermo Fisher Scientific) and allophycocyanin (APC)-conjugated mouse or rat anti-FOXP3 (FJK-16s) antibody (Thermo Fisher Scientific) for 1 h at 25 °C. The sections were then incubated with Hoechst 33,258 (“Molecular Probes, Eugene, OR, USA) for 10 min at 25 °C, for nuclear staining. The specimens were mounted on glass slides using Fluoromount (“Diagnostic BioSystems, Pleasanton, CA, USA) and analyzed and imaged using the BZ9000 fluorescence microscope (Keyence, Osaka, Japan).

### 2.6. Hematoxylin and Eosin (H&E) Staining

To assess the adverse effects associated with AR, nasal epithelial tissues of mice injected with FOXP3 or FOXP3-9R proteins 24 h before OVA administration were stained using H&E 3 h after OVA administration. A BZ9000 fluorescence microscope was used to analyze and image the samples (Keyence).

### 2.7. Antibody Staining and Flow Cytometry

To examine Treg expression, nasal and spleen cells were extracted from treated mice by gentle processing between the ends of two sterile frosted slides. Cell surface staining and flow cytometric analysis of FITC-conjugated mouse anti-CD4 (RM4-5), PE-conjugated mouse anti-CD25 (PC61.5), and APC-conjugated mouse or rat anti-FOXP3 (FJK-16s) expression were performed using the eBioscience FOXP3 “Mouse Regulatory T cell Staining Kit #2” (Thermo Fisher Scientific) as previously described [24]. FITC-conjugated CD4^+^ T cells and APC-conjugated FOXP3^+^ T cells were positively selected using the CytoFLEX^®^ flow cytometer (Beckman Coulter, Brea, CA, USA) to isolate Tregs. The purity of sorted populations was estimated to be >99%. Fluorescence-activated cell sorting data were analyzed using the Kaluza 2.1 software package (Beckman Coulter) [25].

### 2.8. Dot Blotting

To examine the expression of cytokines, dot blot analysis was performed using a mouse cytokine antibody array kit (Membrane, 22 Targets, ab133993; Abcam, Cambridge, UK) following the manufacturer’s instructions. Briefly, nasal epithelial tissues were homogenized and sonicated for 10 min to produce a uniform protein suspension and diluted in concentrations of 250 µg/mL using a BCA protein assay kit (Thermo Fisher Scientific) as previously described [26]. After mixing, 1 mL aliquots were spotted onto membranes and incubated for 2 h at 25 °C. These blots were rinsed and incubated in washing buffer. The blots were then subjected to biotin–streptavidin labeling and detection using horseradish peroxidase-conjugated antibodies. Blots were dried, wrapped in plastic film, and imaged using the “LAS-3000 imaging system (FujiFilm, Tokyo, Japan). ImageJ 1.50i software” (National Institutes of Health, Bethesda, MD, USA) was used to analyze the results as previously reported [27,28].

### 2.9. Statistical Analysis

GraphPad Prism version 9.50 for Windows (GraphPad Software, San Diego, CA, USA) was utilized for all statistical analyses. Values are expressed as the mean ± standard error. Repeated measurements by two-way ANOVA with Tukey’s post hoc correction were used to determine variations among groups. *p*-values < 0.05 were considered statistically significant.

## 3. Results

### 3.1. Nasal Symptoms after Administration of OVA in OVA-IgE Mice

After OVA administration, nasal itching in OVA-IgE mice was monitored at each time point. Nasal symptoms of AR began 1 h after administration, peaked at 3 h, and subsided by 48 h (Figure 2, Saline condition).

### 3.2. Effect of FOXP3-9R on Nasal Symptoms

We assessed whether the injection of FOXP3-9R or FOXP3 to mouse nasal mucosa affected nasal symptoms. A FOXP3-9R injection 24 h before OVA administration was the most effective in suppressing nasal itching, followed by a FOXP3-9R injection 96 h before OVA administration and FOXP3-9R and FOXP3 immediately after OVA administration (Figure 2).

### 3.3. Histopathological Changes

We performed H&E staining of the nasal epithelium of mice receiving FOXP3-9R or FOXP3 injection 24 h before OVA administration. Mice in the *FOXP3 24 h before OVA* and *saline* groups showed mucosal edema or eosinophil infiltration and dilated secretory ducts of the lamina (Figure 3). Conversely, the pseudostratified columnar epithelium structure was normal in the *FOXP3-9R 24 h before OVA* group; in this group, kinocilia were uniform on the epithelial surface. Moreover, the secretory ducts of the lamina propria were not dilated, and there was no evidence of mucosal edema or eosinophil infiltration in this group of animals (Figure 3).

### 3.4. Flow Cytometric Analysis and Immunohistochemistry

We conducted a flow cytometric analysis of the nasal epithelium from OVA-IgE mice after OVA administration. The CD4^+^ FOXP3^+^ cell numbers in the nasal epithelium were considerably different between mice receiving FOXP3 and FOXP3-9R 24 h before OVA administration and immediately after OVA administration (24 h before administration: 3.7% and 6.2%, respectively, *p* < 0.001, Figure 4a,b; immediately after administration: 2.9% and 4.5%, respectively, *p* < 0.001, Figure 4b). However, no major differences were detected when the proteins were administered 96 h before OVA administration (0.8% and 1.2%, respectively, *p* = 0.48, Figure 4b). After FOXP3-9R administration, CD4^+^ FOXP3^+^ cell numbers were the highest in the *FOXP3-9R 24 h before OVA* group (Figure 4a,b). The frozen nasal epithelium sections were immunostained with the appropriate antibodies to analyze the expression of CD4^+^ FOXP3^+^ cells. The results showed that CD4^+^ FOXP3^+^ cells congregated in the lamina propria in the *FOXP3-9R 24 h before OVA* group but not in the *FOXP3 24 h before OVA* group (Figure 4c). CD4^+^ FOXP3^+^ cells were not found in the brain of the *FOXP3-9R 24 h before OVA* group (Appendix A).

### 3.5. Cytokine Analysis

Nasal epithelium was isolated from the six experimental groups for performing bioassays; different ratios of Treg and CD4^+^ Path cells were analyzed for proinflammatory cytokines. CD4^+^ Path T cells in mice that received FOXP3-9R 24 h before OVA had lower levels of interleukin (IL)-4, IL-5, and IL-17 than in those that received FOXP3 24 h before OVA (Figure 5a,b and Appendix A, IL-4, *p* = 0.002; IL-5, *p* = 0.03; IL-17, *p* < 0.001). Regarding the time course, compared to mice receiving FOXP3 immediately after OVA administration, mice receiving FOXP3-9R 24 h before OVA administration had lower levels of IL-4, IL-5, and IL-17 and higher levels of IL-9 and IL-13 in their CD4^+^ Path T cells (Figure 5a,c and Appendix A, IL-4, *p* < 0.001; IL-5, *p* = 0.005; IL-17, *p* = 0.002; IL-9, *p* < 0.001; IL-13, *p* = 0.001). Moreover, mice receiving FOXP3-9R 96 h before OVA administration had lower levels of IL-2, IL-9, IL-10, IL-12p70, and IL-13 and higher levels of IL-4 and IL-5 than those receiving FOXP3 24 h before OVA administration (Figure 5a,c, IL-2, *p* = 0.04; IL-9, *p* < 0.001; IL-10, *p* = 0.01; IL-12p70, *p* = 0.003; IL-13, *p* < 0.001; IL-4, *p* = 0.01; IL-5, *p* = 0.02).

## 4. Discussion

This study demonstrated that transduction of polyarginine-fused recombinant FOXP3 into the nasal epithelium of OVA-IgE transgenic mice may induce immune tolerance via homing of Tregs and activation of Th2, thereby preventing AR-related nasal symptoms. Administration of FOXP3 without polyarginine resulted in partial transduction; however, it did not alleviate the nasal symptoms. Similar to findings in previous studies, mice that underwent 9R-mediated protein transduction did not exhibit deterioration in terms of nasal symptoms or histopathological findings. In our study, AR-related histopathological findings, such as mucosal edema, increased inflammation, cilia loss, vascular dilatation, glandular hyperplasia, eosinophil infiltration in lamina propria, and dilatation and vacuolization in the pseudostratified columnar epithelium on the nasal mucosa surface, were observed in the FOXP3 and saline groups but not in the FOXP3-9R groups. Further, FOXP3-9R mice that maintained nasal mucosal integrity did not exhibit local side effects such as epithelial necrosis, irritation, or hemorrhage. Notably, the histopathological findings were substantially different between mice receiving FOXP3-9R and FOXP3. CPPs help cargo molecules penetrate cells and preserve their biological activity [8,11]. Therefore, another possible use of this approach is the delivery of CPP-mediated oligonucleotides for RNA-based gene silencing to suppress or inhibit gene expression [29]. Consequently, CPPs may serve as effective therapeutic tools against various human pathologies.

FOXP3 transduction can trigger immune tolerance against AR by inducing Treg-like cell production. FOXP3 also contributes to the maintenance of autoimmune system homeostasis. For example, extracellular adenosine is transported into the cytoplasm more easily by FOXP3 [30]. In our study, immune tolerance was likely the result of a SLIT response, which leads to anti-allergen tolerance, possibly through the reprogramming of allergen-specific Th2 cells into Th1 cells and the generation of peripheral Tregs; additionally, the SLIT response suppresses ILs [16,31], which was also observed in this study. These findings suggest that despite the involvement of inflammatory cytokines in FOXP3 transcriptional repression, signals mediated by T cell receptors and cytokine receptors enhance the suppressive activity of Tregs to prevent inflammation after an inflammatory challenge. Recent research has established the existence of finely and timely regulated systems that control the role of Tregs during an inflammatory response by demonstrating that activation- and inflammation-induced alterations in experienced Tregs are lost over time to prevent overall immunosuppression. To date, there are only a few reports regarding the use of Tregs in allergy therapy. Xu et al. reported that hydrogen-rich saline induces Tregs and alleviates AR [32]. Recently, cell-based therapy such as adoptive cell transfer of engineered Tregs was found to provide an effective therapeutic alternative to combat autoimmune diseases [33]. However, these previous studies did not specifically address the role of the *FOXP3* gene or FOXP3 protein.

The use of Tregs in therapeutic applications may involve risks, as Tregs transferred to patients can be altered into Th17 cells that lack regulatory functions [34]; Th17 cells are proinflammatory cells formed in environments similar to those of Tregs [34]. In addition, the use of Tregs in therapy can generate tumors through the action of FOXP3 [35,36]. Our study used locally administered FOXP3, resulting in safer outcomes than those obtained with systemic administration. In addition, no tumorigenesis was observed in our experiment. Thus, our findings suggest that the induction of Tregs via FOXP3 activation and the local administration of FOXP3 before the onset of nasal symptoms have potential for AR therapy applications. Our study provides insights into putative novel AR-therapeutic approaches.

This study had several limitations. First, we did not monitor any long-term effects. As previously described for systemic administration, long-term observation may reveal tumorigenesis, off-target effects, and genesis of chronic inflammation through changing the T cell phenotype by immune responses [37]. In addition, SLIT typically warrants long-term therapy. Compared to long-term SLIT, our study might be too brief for acquiring immune tolerance. In a previous study using Balb/cJ mice, SLIT was performed 5 days a week for 6–9 weeks [38]. They showed that the effect of SLIT is time-dependent. Treatment for 9 weeks was able to ameliorate both clinical symptoms, eosinophilia, allergen-specific IgE as well as the local T cell response. A shorter treatment period still had an effect on the levels of IgE, whereas there was no effect on the clinical symptoms [38]. Such time-dependency has also been observed in humans. In a large, tablet-based SLIT study, it was shown that 8 weeks or more of pre-seasonal treatment was significantly more effective than 6 weeks, with respect to reduction in symptom score and medication requirements [39]. Second, our therapeutic approach focused on prevention and not the treatment of existing disease conditions. Third, the mechanisms for induction of Tregs were unknown in our study. Experimental evidence has shown that the immune system can induce peripheral mechanisms of immune tolerance to allergens [40]. The generation of Tregs can be influenced by factors such as FOXP3+ Treg, pathogen-derived molecules, and exogenous signals such as histamine, adenosine, vitamin D3 metabolites, or retinoic acid [40,41]. While the molecular mechanisms of Treg generation in vivo are not fully understood, recent studies have provided insights into these processes. There is a counter-regulation between Th2 and Treg responses in healthy individuals and allergy patients. The transcription factor GATA3 directly inhibits the expression of FOXP3, hindering tolerance induction by Th2-type immune responses. In autoimmune disease models, a dichotomy between pathogenic Th17 and protective Treg responses has been observed, with TGF-β contributing to the generation of both. The presence of IL-6 shifts the balance towards Th17 generation [42,43]. Retinoic acid also influences the balance between inflammatory Th17 cells and suppressive Tregs by inhibiting Th17 formation and enhancing FOXP3 expression through a signaling pathway independent of STAT3/STAT5 [41,44]. Overall, further analysis of long-term observations, side effects, and drug function enhancement is required for the clinical application of our therapeutic strategy.

## 5. Conclusions

Our study demonstrated that effective intranasal FOXP3-9R delivery into the nasal mucosa of mice induced Treg-like cells and generated anti-AR tolerance. Furthermore, cytokine assays indicated that FOXP3-9R transduction before OVA administration induced a SLIT-like response in the mouse nasal epithelium. These results suggest that CPP-mediated local protein transduction and immune tolerance therapy are potential alternatives to conventional protein transduction methods for the delivery of therapeutically relevant molecules for AR therapy. These findings shed light on new AR therapeutic approaches. However, further research is warranted for the definitive establishment of AR therapy using this strategy.

## Figures and Tables

**Figure 1 pharmaceutics-15-01770-f001:**
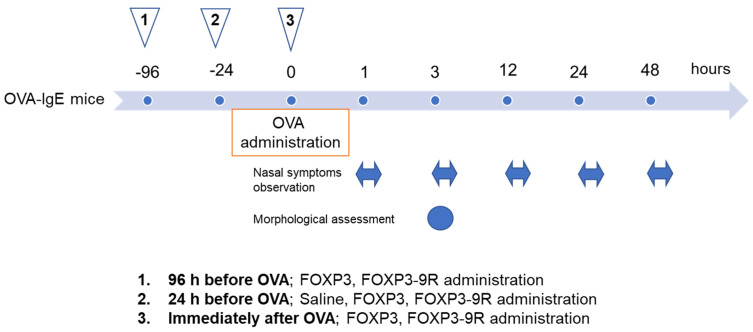
Study design. FOXP3, FOXP3-9R, or saline were administered into the nasal cavities of OVA-IgE mice. Triangle 1 indicates FOXP3 or FOXP3-9R injection 96 h before OVA administration; Triangle 2 indicates FOXP3, FOXP3-9R, or saline injection 24 h before OVA administration; and Triangle 3 indicates FOXP3 or FOXP3-9R injection immediately after OVA administration. Nasal symptoms were observed at 1, 3, 12, 24, and 48 h after OVA administration, for 10 min (bilateral arrows). Morphological assessments were performed 3 h after OVA administration.

**Figure 2 pharmaceutics-15-01770-f002:**
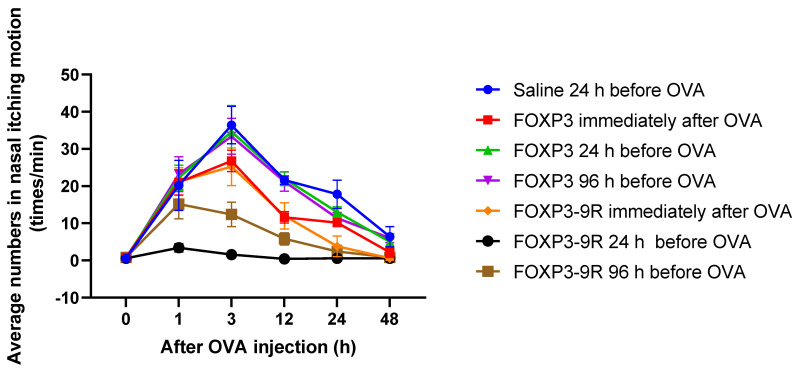
Assessment of nasal symptoms. The number of nasal itching motions (times/min) was counted using animal video files.

**Figure 3 pharmaceutics-15-01770-f003:**
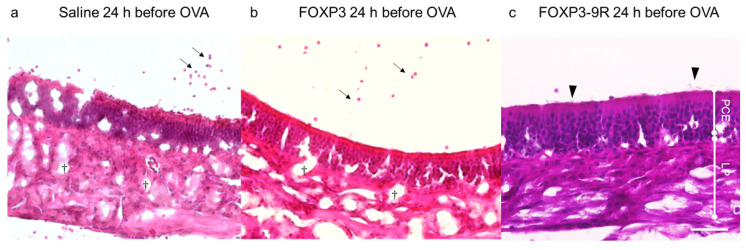
Histopathological findings in the nasal epithelium. H&E staining was performed 3 h after OVA administration. (**a**–**c**) Representative images of the corresponding experimental conditions: (**a**) Saline injected 24 h before OVA administration. (**b**) FOXP3 injected 24 h before OVA administration. (**c**) FOXP3-9R injected 24 h before OVA administration. H&E, hematoxylin and eosin stain; LP, lamina propria; PCE, pseudostratified ciliated columnar epithelium. Arrows indicate eosinophil infiltration. Daggers indicate dilated secretory ducts. Arrow heads indicate kinocilia. Scale bar: 50 µm.

**Figure 4 pharmaceutics-15-01770-f004:**
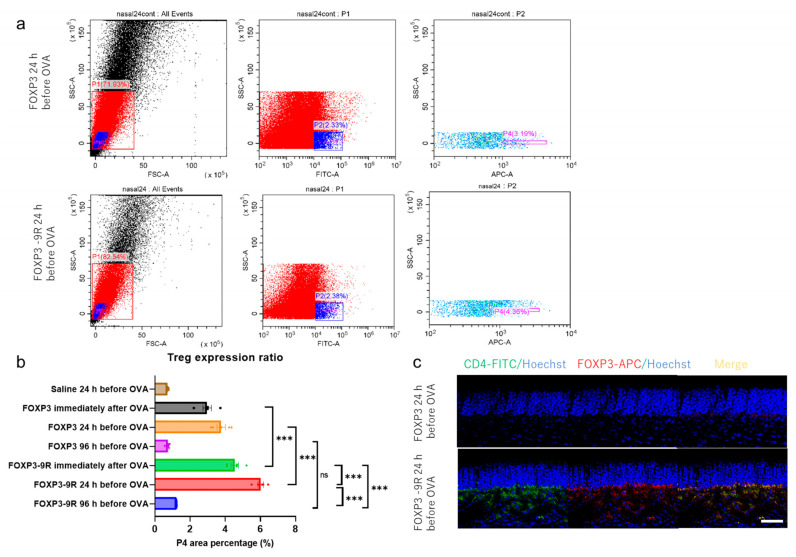
Flow cytometric analysis and immunohistochemistry. (**a**) Treg-like cells expressed in the nasal epithelium. FITC and APC were conjugated to CD4 and FOXP3, respectively. The P4 population was indicated in the CD4^+^ FOXP3^+^ cells (Treg-like cells). (**b**) The P4 area percentage in each treatment group. Statistical comparisons between the FOXP3 and FOXP3-9R groups and among FOXP3-9R groups alone are depicted in the graph. (**c**) Immunohistochemistry findings. Green, red, and blue indicate CD4-FITC, FOXP3-APC, and Hoechst staining, respectively. ***: *p* < 0.001. ns = not significant. Scale bar: 50 µm.

**Figure 5 pharmaceutics-15-01770-f005:**
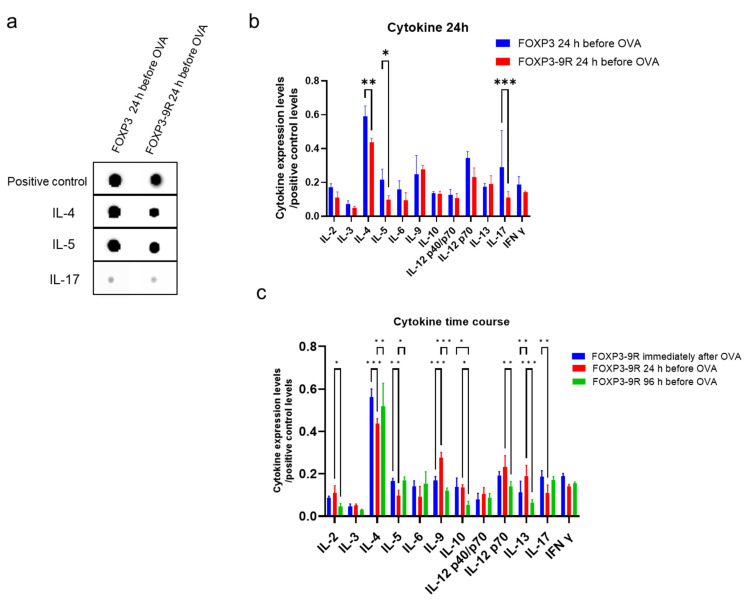
Cytokine assay. (**a**) Dot blot assay. (**b**) Comparison between FOXP3 and FOXP3-9R 24 h before OVA groups. (**c**) Changes in cytokine expression levels among FOXP3-9R groups. *: *p* < 0.05, **: *p* < 0.01, ***: *p* < 0.001.

## Data Availability

All data will be provided upon reasonable request.

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
