# Peer review of "Arginine-Rich Cell-Penetrating Peptide-Mediated Transduction of Mouse Nasal Cells with FOXP3 Protein Alleviates Allergic Rhinitis"

_pharmaceutics, 2023, doi:10.3390/pharmaceutics15061770_

Round 1

Reviewer 1 Report

Dear colleagues ,

congratulation to your manuscript. It presents a possible way for treatment of allergic rhinitis, describes the molecular and cellular mechanisms of action. I would like to emphasize that this manuscript also discusses possible side effects by longtime teatment evoked by changing the  phenotype of T-cells.

One remark: Not only cationic peptides belong to the group of cell-penetrating peptides, but also amphipatic and even anionic peptides. Depending on cell line some of them are more efficient than R9.  Figure 5: Please explain in legends the empty bars with black lines.  Thank you for the supplement. It gives a good impression about the test and supports the very interesting results with a great variety of inflammary cytokines.  The references are correct and necessary, but in many cases more than one decade old.  Do you have more recent ones?

Author Response

congratulation to your manuscript. It presents a possible way for treatment of allergic rhinitis, describes the molecular and cellular mechanisms of action. I would like to emphasize that this manuscript also discusses possible side effects by longtime treatment evoked by changing the phenotype of T-cells.

> Thank you.

One remark: Not only cationic peptides belong to the group of cell-penetrating peptides, but also amphipatic and even anionic peptides. Depending on cell line some of them are more efficient than R9. 

>Thank you for highlighting this. I have changed the definition and description of CPPs.

Figure 5: Please explain in legends the empty bars with black lines.  Thank you for the supplement. It gives a good impression about the test and supports the very interesting results with a great variety of inflammary cytokines. 

> The empty black bars indicate statistical comparisons by ANOVA.

The references are correct and necessary, but in many cases more than one decade old.  Do you have more recent ones?

> Thank you for highlighting this. I have revised them.

Reviewer 2 Report

"Arginine-rich cell-penetrating peptide-mediated transduction of mouse nasal cells with FOXP3 protein alleviates allergic rhinitis" is an interesting paper by Maiwa and colleagues. Although this would be of great, general interest to our readership, this reviewer has a few questions/clarifications requested.

1). Under Immunohistochemistry, authors state: "In groups injected with FOXP3-9R and FOXP3 24 h after OVA administration, a total of 10 mice (n = 5 per time point) were euthanized 3 h after OVA administration". This is confusing. Is this a mistake. None of the mice had FOXP3 or FOXP3-9R administered AFTER the OVA administration. All had the two administered either before or AT THE TIME of OVA administration as per Figure 1. Please clarify.

2). There is a use of double negatives in this sentence which is confusing. "However, no studies have not reported directly on the application of the Foxp3 gene or protein 63 for AR therapy". 

3). How do the authors explain the effect of FOXP3 and FOXP3-9R administered immediately after OVA administration being nearly identical. Similarly, why is FOXP3 alone having biological effects as seen with upregulation of Treg expression (Figure 4B)? Is the protein being taken up, albeit lower levels, without presence of the arginine homopolymer?

4). The H&E staining in Figure 3 is low magnification/low resolution, and not high enough quality to demonstrate the points being made. Arrows and better labeling would also help.

5). Similarly, the immunohistochemistry panels in Figure 3C are low magnification/resolution and difficult to understand-better labeling necessary. 

6). An important control group (saline 24 hrs before followed by OVA administration) is missing from Figure 5. That data needs to be added to the Figure 5 to show basal degree of activation with OVA alone administration. 

7). Lastly, the authors state in the Discussion section "Third, we did not examine adverse effects on brain function; nevertheless, no aberrant 292 behavior was observed in the mice". This is an over-reach as authors did not do formal behavioral testing. Additionally, did they look for any uptake in brain tissue of FOXP3-9R? Was there any staining observed of brain tissue. This is an important negative that should be included. 

Minor English editing needed. See comment 2 above. 

Author Response

"Arginine-rich cell-penetrating peptide-mediated transduction of mouse nasal cells with FOXP3 protein alleviates allergic rhinitis" is an interesting paper by Maiwa and colleagues. Although this would be of great, general interest to our readership, this reviewer has a few questions/clarifications requested.

1). Under Immunohistochemistry, authors state: "In groups injected with FOXP3-9R and FOXP3 24 h after OVA administration, a total of 10 mice (n = 5 per time point) were euthanized 3 h after OVA administration". This is confusing. Is this a mistake. None of the mice had FOXP3 or FOXP3-9R administered AFTER the OVA administration. All had the two administered either before or AT THE TIME of OVA administration as per Figure 1. Please clarify.

>Thank you for pointing out this error. It has been revised from “after” to “before.”

2). There is a use of double negatives in this sentence which is confusing. "However, no studies have not reported directly on the application of the Foxp3 gene or protein 63 for AR therapy".

>This has been revised as follows: “no studies have reported directly on the application of the Foxp3 gene or protein for treating AR.”

3). How do the authors explain the effect of FOXP3 and FOXP3-9R administered immediately after OVA administration being nearly identical. Similarly, why is FOXP3 alone having biological effects as seen with upregulation of Treg expression (Figure 4B)? Is the protein being taken up, albeit lower levels, without presence of the arginine homopolymer?

>Thank you for this comment. We believe that Foxp3 alone can be introduced into the nasal mucosa to some extent. However, considering the transfection efficiency, duration of expression, and peak expression level after transfection, the effect of Foxp3 alone is expected to be considerably lower than that of Foxp3-9R.

4). The H&E staining in Figure 3 is low magnification/low resolution, and not high enough quality to demonstrate the points being made. Arrows and better labeling would also help.

>Thank you for this valuable comment. The image of Figure 3C has been changed with greater magnification and improved resolution and arrows have been added to the images.

5). Similarly, the immunohistochemistry panels in Figure 3C are low magnification/resolution and difficult to understand-better labeling necessary.

> As suggested, the image of Figure 3C has been changed.

6). An important control group (saline 24 hrs before followed by OVA administration) is missing from Figure 5. That data needs to be added to the Figure 5 to show basal degree of activation with OVA alone administration.

>Thank you for raising this point. Unfortunately, we did not perform assays with the saline 24 h control owing to lack of funds. Previous studies have reported that levels of interleukins (ILs) such as IL-4, IL-5, and IL-17 are elevated in allergic rhinitis (AR) responses. Our results demonstrate that these ILs are suppressed by the described treatment; therefore, we considered that our therapy improved AR.

7). Lastly, the authors state in the Discussion section "Third, we did not examine adverse effects on brain function; nevertheless, no aberrant 292 behavior was observed in the mice". This is an over-reach as authors did not do formal behavioral testing. Additionally, did they look for any uptake in brain tissue of FOXP3-9R? Was there any staining observed of brain tissue. This is an important negative that should be included.

>This description has been removed to avoid an overstatement. In addition, brain tissue staining image has been added to Supplementary Figure 2.

Reviewer 3 Report

This paper focuses on the use of the FOXP3-9R hybrid protein as a new strategy against allergic rhinitis. The authors propose the use of polyarginine (9R) as a means to transduce FOXP3 protein into mouse nasal cells. The method seems ineteresting, although it is quite widely used in biochemical and pharmacological studies. Its novelty is linked to the use of the FOXP3-9R hybrid in the search for new pharmacological strategies for the treatment of allergic rhinitis. In principle, I have no comments on the study methodology presented by the authors and the description of the results obtained. 

My main reservation and defect of the manuscript is the inadequate characterization of the title FOXP3 protein. Apart from the information in section 2.1 that the authors obtained a hybrid recombinant protein, there is virtually no information about it.  

The authors do not clarify in the paper whether the notation FOXP3-9R means that the CPP peptide was attached to the C-terminus of the protein (FOXP3), as the nomenclature of this notation would indicate. The literature reference [16] does not clarify this either.

It would be useful to include any characterization of FOXP3-9R in the supplement section, e.g. SDS PAGE. 

Section 2.1. What is the concentration (stock solution) of FOXP3 protein and hybrid with 9R after dialysis?

section 2.3 s ...(10 µL aliquots) were administered.... Without stating the concentration of this solution, such information is worthless. The authors do not explain why they chose this particular amount of protein. Was the concentration/quantity of protein optimized?

Figure 4 part a. The description of the figures is too small and thus illegible.  

Author Response

This paper focuses on the use of the FOXP3-9R hybrid protein as a new strategy against allergic rhinitis. The authors propose the use of polyarginine (9R) as a means to transduce FOXP3 protein into mouse nasal cells. The method seems ineteresting, although it is quite widely used in biochemical and pharmacological studies. Its novelty is linked to the use of the FOXP3-9R hybrid in the search for new pharmacological strategies for the treatment of allergic rhinitis. In principle, I have no comments on the study methodology presented by the authors and the description of the results obtained.

My main reservation and defect of the manuscript is the inadequate characterization of the title FOXP3 protein. Apart from the information in section 2.1 that the authors obtained a hybrid recombinant protein, there is virtually no information about it. 

>Thank you for this valuable comment. Accordingly, information about the FOXP3-9R protein has been added to the revised manuscript.

The authors do not clarify in the paper whether the notation FOXP3-9R means that the CPP peptide was attached to the C-terminus of the protein (FOXP3), as the nomenclature of this notation would indicate. The literature reference [16] does not clarify this either.

>Thank you for pointing this out. The sequence file has now been added to Supplementary Figure 1a. Further, the construction of FOXP3-9R, with the 9R peptide attached to the C-terminus of the protein, has been described in the revised version.

It would be useful to include any characterization of FOXP3-9R in the supplement section, e.g. SDS PAGE.

>Thank you for this suggestion. The SDS-PAGE results for FOXP3-9R characterization have been added to Supplementary Figure 1b.

Section 2.1. What is the concentration (stock solution) of FOXP3 protein and hybrid with 9R after dialysis?

>The concentration of FOXP3 and FOXP3-9R was 0.3 mg/mL and 0.2 mg/mL, respectively. This has been described in the Methods section of the revised paper.

section 2.3 s ...(10 µL aliquots) were administered.... Without stating the concentration of this solution, such information is worthless. The authors do not explain why they chose this particular amount of protein. Was the concentration/quantity of protein optimized?

>Thank you for this comment. The protein concentration was indicated above. The proteins were diluted in PBS to the same concentration (0.1 mg/mL) and the same amounts (10 µL aliquots) were used. This description has been added to the Methods section.

Figure 4 part a. The description of the figures is too small and thus illegible. 

>Thank you for pointing this out. The figure description has been enlarged for improved readability.

Round 2

Reviewer 2 Report

The authors have addressed all the issues this Reviewer raised.

Minor spell/grammar check only required now before publication.